# Genome-wide analysis of a collective grave from Mentesh Tepe provides insight into the population structure of early neolithic population in the South Caucasus

Perle Guarino-Vignon [1,2 ✉], Maël Lefeuvre[1], Amélie Chimènes[1], Aurore Monnereau[3], Farhad Guliyev[4], Laure Pecqueur[5,1], Elsa Jovenet[5], Bertille Lyonnet[6] & Céline Bon [1 ✉]

Despite the localisation of the southern Caucasus at the outskirt of the Fertile Crescent, the Neolithisation process started there only at the beginning of the sixth millennium with the Shomutepe-Shulaveri culture of yet unclear origins. We present here genomic data for three new individuals from Mentesh Tepe in Azerbaijan, dating back to the beginnings of the Shomutepe-Shulaveri culture. We evidence that two juveniles, buried embracing each other, were brothers. We show that the Mentesh Tepe Neolithic population is the product of a recent gene flow between the Anatolian farmer-related population and the Caucasus/Iranian population, demonstrating that population admixture was at the core of the development of agriculture in the South Caucasus. By comparing Bronze Age individuals from the South Caucasus with Neolithic individuals from the same region, including Mentesh Tepe, we evidence that gene flows between Pontic Steppe populations and Mentesh Tepe-related groups contributed to the makeup of the Late Bronze Age and modern Caucasian populations. Our results show that the high cultural diversity during the Neolithic period of the South Caucasus deserves close genetic analysis.

[1] UMR7206 Éco-Anthropologie (EA), CNRS, Muséum National d'Histoire Naturelle, Université Paris-Cité, Paris, France. [2] UMR5288 CAGT, CNRS, Université Paul Sabatier, Toulouse, France. [3] Department of Archaeology, University of York, BioArCh, Environment Building Wentworth Way Heslington, York YO10 5NG, UK. [4] Head of the Science Fund and the Museum Department of the Institute of Archeology, Ethnography and Anthropology, Azerbaijan National Academy of Science, Baku, Azerbaijan. [5] Institut National de Recherches Archéologiques Préventives, Centre-île de France, Paris, France. [6] UMR7192 PROCLAC « Proche-Orient – Caucase: langues, archéologie, cultures », CNRS, Paris, France. ✉email: perle.gv@gmail.com; celine.bon@mnhn.fr

In the Near East, the Neolithic way of life emerged between 9000 and 7000 BCE. Several centres of Neolithisation have been identified, such as the Levant or Southern China, from which the agropastoral way of life diffused to other regions. The mechanisms of this diffusion have attracted tremendous attention for the last few decades. In some places, the Neolithic gained ground through the acculturation of local hunter-gatherers (for instance, in Anatolia or Iran);[1] but in most regions (Europe, South-East Asia), farmer populations spread, and assimilation processes took place with a degree of admixture (for a review, see ref. [2]).

The mechanism of Neolithisation in the South Caucasus, a region located between the Black and Caspian Seas on the southern slope of the Greater Caucasus Mountains, remains poorly understood. Mesolithic sites are known at Damjili Cave, unit 5 (Western Azerbaijan)[3], Kmlo-2 Rock Shelter (Western Armenia) and Kotias Klde Cave (Western Georgia). Paleogenetic analyses of human bones excavated from the Kotias Klde Cave showed a genetic continuity with earlier Upper Palaeolithic (post-Last Glacial Maximum/LGM) sites[4] but a discontinuity with pre-LGM individuals[5]. Their genetic ancestry shares, to a certain extent, a common origin with ancient Iranian populations[6] and differs from that of Anatolian and Levant hunter-gatherer groups[1,7], demonstrating a high genetic differentiation at this time between geographically close populations[8].

The first settlements attributed to the Early Neolithic period belong to an aceramic culture, evidenced in several places in Central Georgia, as at Nagutni, in Western Georgia, as at Paluri[9] and in Western Azerbaijan at Damjili Cave, unit 4[3]. However, evidence of agriculture and herding remains scarce, suggesting that these sites represent a transitional phase between the Mesolithic and the Neolithic.

In this context, the Shomutepe-Shulaveri culture (SSC) is the most ancient Caucasus culture with a complete Neolithic package[9–11]. Found in several clusters of settlements in the northern foothills of the lesser Caucasus, the SSC is characterised by circular mud-brick houses, domestic animals and cereals, handmade pottery, sometimes with incised and relief decoration, and obsidian and bone industries. Variants, such as the Aratashen/

Aknashen culture[9,12] (Ararat Plain) and other Neolithic contemporaneous cultures like the Kültepe[13] Culture (Nakhchivan region) are also found in the South Caucasus. The slightly later Kamiltepe culture (Mil steppe culture), which probably includes the site of Polutepe, differs by its architecture, the use of flint tools instead of obsidian, and pottery-painted patterns that are rather related to Northern Iran and the Zagros[14,15].

The origins of the SSC are still discussed[10,11]. Due to the rapid transition from the aceramic stage to the SSC, population continuity during the Neolithisation process is possible[3,12]. However, several cultural and biological features are nonlocal. Domesticated animals, such as cattle, pigs or goats, originate from Eastern Anatolia and from the Zagros moutains[16–18]. Similarly, the glume wheat and barley recovered in the SSC sites have been domesticated elsewhere in the Middle East[19], even if the ancestors of the naked wheat *Aegilops tauschii* are found in the Caucasus and may have been local[20]. The material culture and architecture evidence technical transfers with neighbouring regions such as Southern Anatolia, Pre-Halafian and Halafian culture of Northern Mesopotamia and Zagros[11,21]. Taken together, these data suggest a strong cultural connection, and maybe a degree of admixture, with other groups from the Fertile Crescent. Indeed, the genome-wide data for one individual coming from the same collective grave at Mentesh Tepe as the samples we analyse here, and confirmed by that of other individuals from Polutepe in the Mughan steppe or from Aknashen and Masis Blur in Armenia[22] already showed that southern Caucasian groups are part of a cline that connected Eastern Anatolian and Zagros populations and evidence a gene flow that began around 6500 years BC[23].

One of the oldest sites of the SSC, Mentesh Tepe, is located in the Tovuz district of western Azerbaijan and has been excavated between 2007 and 2015 (Fig. 1a). Several occupations were revealed, the earliest dating back to the Neolithic SSC period[24,25]. At this time, the botanical assemblage is dominated by cereals, especially barley, naked wheat and emmer[26], a common association during the Neolithic Southern Caucasus. Animal remains consist largely of domesticated ones (ovicaprines, cattle, pigs, dogs), and wild animals are rare[27]. The diet of the Neolithic individuals from Mentesh Tepe relied mainly on C3-plants, such

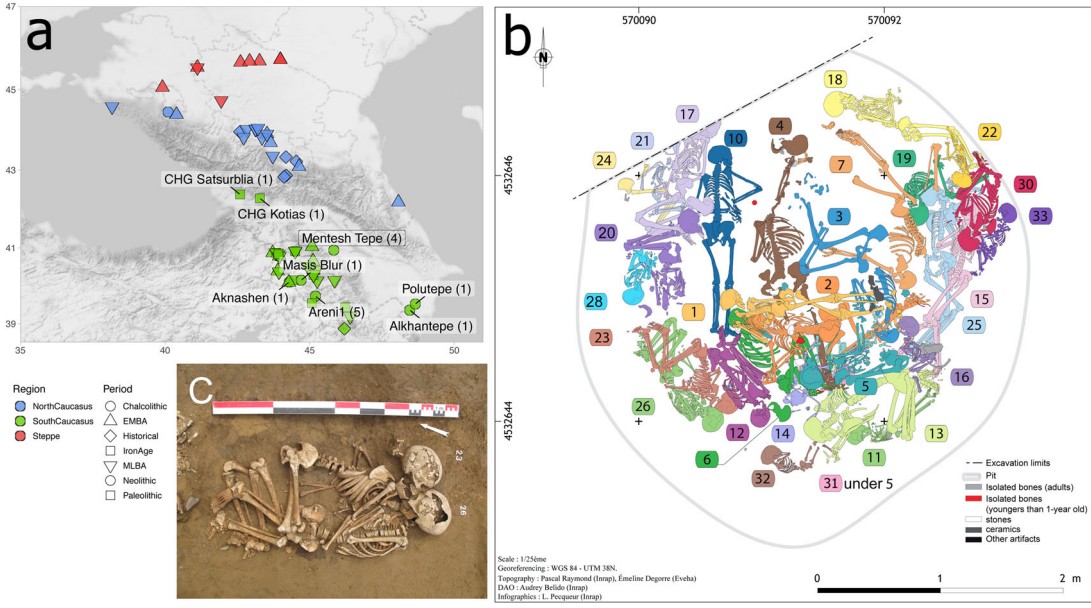

**Fig. 1 Mentesh Tepe. a** Localisation of Mentesh Tepe and other Neolithic and Protohistoric places that provided ancient genomes. Map tiles by Stamen Design, under CC BY 3.0. Data by OpenStreetMap, under ODbL. **b** Structure 342 and localisation of each individual. **c** Photograph of MT23 and MT26 (credit L. Pecqueur).

as wheat, barley and lentils, with some evidence of freshwater fishes; the consumption of animal proteins varies between individuals[28]. The pottery differs from classical SSC sites by being vegetal-tempered, a characteristic shared with Kamiltepe or the first occupation of the Nakhchivan site of Kültepe[29]. As observed in many SSC sites, houses are circular and made of mudbricks, with or without the addition of straw or other organic material[21]. Two SSC occupation phases are represented, separated in some places by a thick layer of ashes.

In a context where Neolithic burials are rare, Mentesh Tepe is exceptional for the discovery of a collective burial containing around 30 individuals (Fig. 1b), which is associated with the end of the first phase of frequentation of the site. Archaeoanthropological analysis has shown that it was a complex funerary gesture with mostly simultaneous deposits and, in contrast, some successive deposits which permitted manipulations on not completely decomposed bodies[30]. The number of individuals in the burial, as well as their sex and age bias, suggest a dramatic event such as an epidemic, a famine, or a sudden episode, but no trace of violence has been evidenced on the bones.

There is no specific orientation or position of the corpses, but some intentional arrangements are visible. The most striking is formed by two juveniles embracing each other (Fig. 1c). Such an arrangement is rare, but other examples have been found in Neolithic and Protohistoric times, such as in Diyarbakir (Turkey, 6100 BC) or Valdaro (Italy, 3000 BC). Double burials are often considered as a lover's embrace, but arguments for this explanation are often elusive[31].

To better understand the origin of the Shomu-Shulaveri population and the structuration of this community, we performed paleogenetic studies of some of the individuals found in the collective burial. The genetic data obtained are then compared to those of another individual from the structure already published and to contemporaneous southwestern Asian genomes.

## Results

**DNA isolation and sequencing.** From the 30 individuals from Structure 342 of the site, we sampled 23 petrous bones but could only obtain genome-wide data for one female (Individual 7, later called MT7) and two males (Individual 23 and Individual 26, MT23 and MT26, respectively), with coverage ranging from 0.1 to 0.3X, a number of SNPs hit on the 1240k from 61,151 to 205,055 and ancient DNA damages (Supplementary Fig. 1). For four other individuals, shallow sequencing and estimation of Rx and Ry only allowed determining biological sex (Supplementary Data 1): two were males and two were females. There is no discrepancy compared to the already published sex determinations made from adult pelvic measurement[30], but genetic data allowed a robust determination of the sex for some juveniles.

**Genetic structure of the Neolithic South-Caucasus.** We merged our genome-wide data with the Human Origins dataset (HO-dataset)[1], as well as with 3529 previously unrelated published ancient genomes (Supplementary Data 2). To decipher the genetic relations between the new Mentesh Tepe individuals and other ancient populations from the Caucasus, Anatolia, the Near East, and the Middle East, we performed: (1) a PCA[32] on the modern dataset on which the ancient genomes have been projected and (2) an unsupervised ADMIXTURE[33] analysis with the HO and the ancient dataset (Fig. 2a, b, Supplementary Fig. 2). The PCA shows that the Mentesh individuals overlap with some other previously published Neolithic or Chalcolithic individuals from the South Caucasus[22,23], but the individual from Aknashen falls a bit closer to CHG than the main neolithic cluster (Fig. 2a), and fall intermediate between the Iran Neolithic cluster and the

Neolithic Anatolian Farmer group. The ADMIXTURE analysis suggests that the Mentesh individuals carry three main components: i.e., ca. 30% Neolithic Iran (Iran_N; green), 15% Levant Neolithic (PPN; pale rose) and 55% blue and pink components shared with Anatolian or European Neolithic populations (Fig. 2b). Both analyses show that the new Mentesh Tepe individuals present similar profile as that already published from individual MTT001 (despite different sequencing strategy) and from other Neolithic (Polutepe, Azerbaijan, Masis Blur and Aknashen, Armenia) or Chalcolithic (Alkhantepe, Azerbaijan) sites in the South Caucasus without true significant variations in percentage (Kruskal–Wallis test, $p$-value = 0.25), even though the Aknashen individual has the highest IranN/CHG percentage. They are also very similar to the Chalcolithic and Bronze Age Anatolian populations from Arslantepe and with other Chalcolithic and Bronze Age Anatolian populations but are quite distant from the Late Neolithic Anatolian Tell Kurdu individuals (five individuals identified as Tell_Kurdu_EC by the original publication, whom we refer as TellKurdu_LN) who instead clusters with the Neolithic Anatolian populations. Due to the lack of genomic data from South-eastern Mesopotamia, only the ancestry of groups that lived in North-eastern Mesopotamia during the PPN period and the Late Bronze Age could be considered. We observe that the South Caucasus population displays a different profile from the PPN Mesopotamian one, as it has more Anatolian affinity, but that it presents a profile similar to the LBA individual from North-eastern Mesopotamia (Nemrik9_LBA).

To formally test for the genetic affinity with earlier populations in Western Eurasia observed in the PCA in our samples, we performed D-statistics of the form D(Mbuti, Y; Z, MT). The statistic deviates significantly from zero if the pair of Anatolian/Caucasian/Mesopotamian groups (Z) and Mentesh (MT) do not have the same genetic relation to Western European populations (Y). We confirm that the new Mentesh samples are very similar to MTT001 as almost no D-statistics of the form D(Mbuti, Z; MTT001, MT23/MT7/MT26) significantly deviate from 0, showing a high level of homogeneity in this site (Supplementary Fig. 3).

When performing the same D-statistic (Fig. 2c) with MT7, MT23 and MTT001 reunited within a single Mentesh Tepe group (but excluding MT26, as he is related to MT23—cf. infra), we observe that they share more alleles with the Early European Farmers (Greece_N, Serbia_EN), Anatolian farmers, Anatolian Epipaleolithic individual (Anatolia EP) and Neolithic Levant (PPN) than with the Caucasus hunter-gatherers (CHG from the Satsurblia and Kotias Klde caves). They also deviate from the Neolithic Anatolian populations (Barcın site in North Anatolia, Tell Kurdu in South-eastern Anatolia and Tepecik Çiftlik in Central Anatolia) by sharing more alleles with CHG and Iran_N (Ganj Dareh in Zagros mountains) but sharing fewer alleles with Western European Hunter-Gatherers (Iberia_HG, Loschbourg, Goyet_Q2, Latvia_HG), Early European Farmers, Anatolia EP, and with the Neolithic Levant. We also note that they do not differ from the Neolithic individuals from Polutepe and from a Late Chalcolithic individual from Alkhantepe, both sites being further east in the Mughan plain of Azerbaijan (Supplementary Fig. 3). They also do not diverge from the Neolithic Armenian groups (Aknashen and Masis Blur). In the end, in regard to the Eurasian diversity, they all form a homogeneous cluster representing a Neolithic ancestry in the South Caucasus, from east to west, sharing a recent ancestry with Neolithic groups found in South-eastern Anatolia.

Using qpAdm[34], we could successfully model the Mentesh Tepe group and each of the individuals as a two-way admixture of TellKurdu_LN, representing South-eastern Anatolian ancestry, and Iran_N ($p$ = 0.20, 37% Iran_N source) (Fig. 2d). Replacing TellKurdu_LN by Barcın_N ($p$ = 0.03–0.75, 33–45% Iran_N

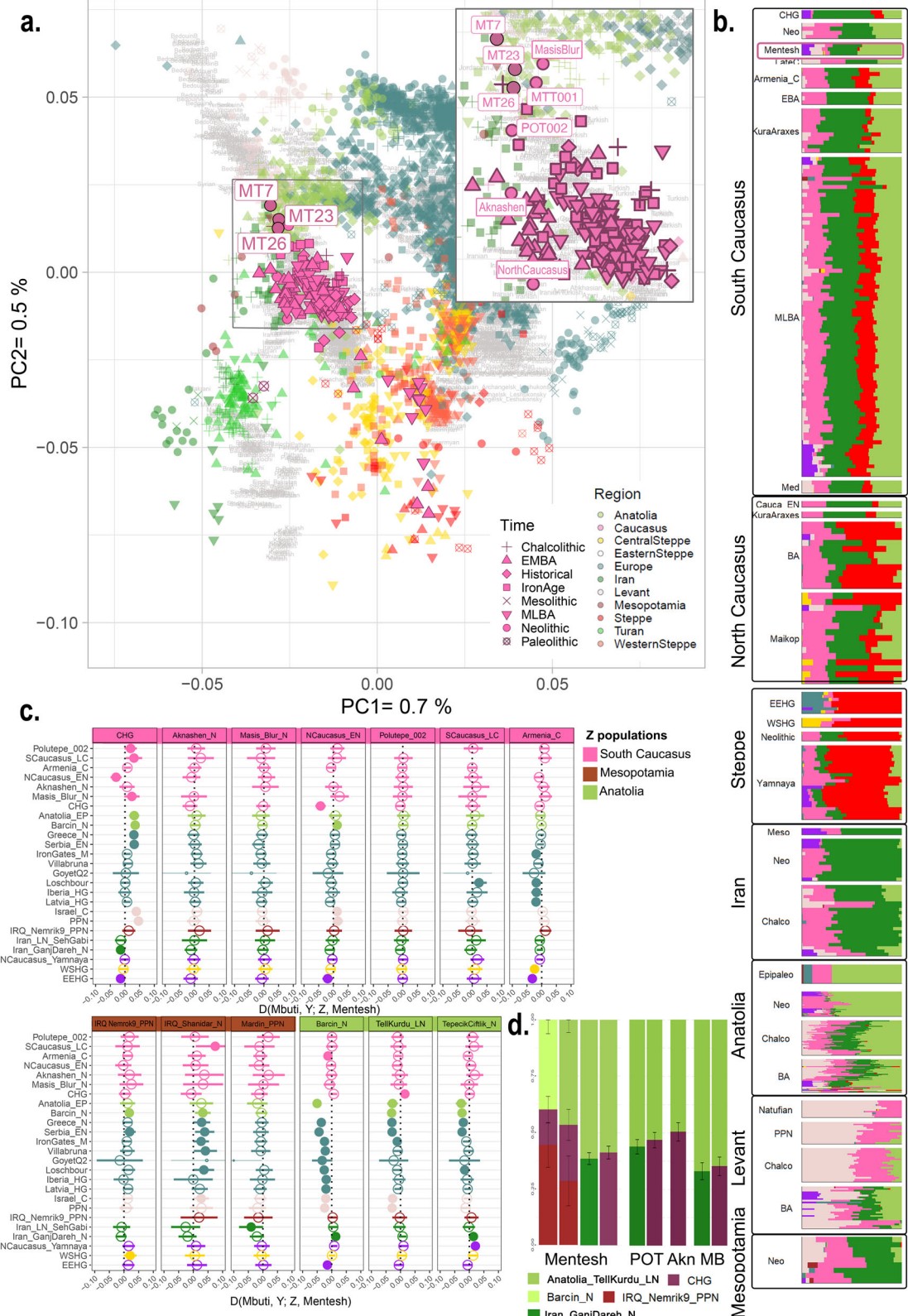

**Fig. 2 Genetic affinities of the Mentesh Tepe individuals. a** Principal Component Analysis calculated on 1390 published present-day individuals (in grey) with ancient individuals projected onto it. The new Mentesh Tepe samples are indicated by filled dark pink circles and are labelled. Zoom panel allows to label all the Caucasus Neolithic individuals. **b** Unsupervised ADMIXTURE analysis, K = 9 for a selection of ancient populations. The Mentesh Tepe individuals appear like other Neolithic and Chalcolithic individuals from the South Caucasus (lowlands_N and LateC). CHG, Caucasus Hunter-Gatherer; Meso, Mesolithic; Levant_EP, Levant Epipaleolithic; C or Chalco, Chalcolithic; EN, Early Neolithic; Neo, Neolithic; BA, Bronze Age. **c** D-statistics D(Mbuti; Y; Z, Mentesh Tepe). D-statistics tests whether either Z (or Mentesh Tepe) has excess affinity with Y and becomes negative (or positive). Values that deviate from 0 by the ±2 Standard Error (SE)—displayed by the error bars—are represented with a filled circle. **d** Ancestry proportions of the Neolithic group on autosomes calculated with qpAdm. Ancestry proportions are plotted with error bars representing ±1 SE. POT = Polutepe, MB = Masis Blur, Akn = Aknashen.

source) and replacing Iran_N by CHG ($p = 0.01–0.45$, 36–49% CHG source) also provide acceptable models for Mentesh as a group but not always for all the samples when modelled individually. We also modelled Mentesh as a mixture between a Neolithic Anatolian group (Tell Kurdu or Barcın) and a North Mesopotamian group (Nemrik9 or Mardin). Eventually, we tested 3-populations models and obtained acceptable models for Mentesh, involving CHG (15–25%), an Anatolian population (Barcın 40% or Tell Kurdu 45%) and a Neolithic North-eastern Mesopotamian population (Nemrik9 29–45%). The same analysis was performed on the other Neolithic South-Caucasus groups (Polutepe, Aknashen and Masis Blur), and we obtained models with a $p > 0.05$, involving the same kind of combinations as for Mentesh. The amount of Anatolian ancestry in these three populations is not significantly different than that in Mentesh Tepe (Wilcoxon test, $p$-value > 0.05 for all three), even though the Aknashen individual presents the highest CHG percentage. (Supplementary Fig. 4).

Eventually, we used DATES[35] to estimate the date of the admixture event. To account for the low number of individuals representing the Neolithic Ganj Dareh in Iran and the poor coverage, we added other individuals from Neolithic sites in Iran (Seh Gabi, Tepe Abdul Hossein, Wezmeh). We found that the admixture between the Anatolian source and the Iranian source only took place 15 ± 5 generations before Mentesh Tepe occupancy ($Z = 2.5$ and $nrmsd = 0.348$). With 28 years per generation, this dates the admixture event around 6300 BC (Phase 1 occupancy: 5880 BC). However, the decay of ancestry covariance estimated by DATES for Mentesh poorly fits with the data (Supplementary Fig. 5).

**Genetic transition to the Bronze Age**. Having established a general profile for the Neolithic South-Caucasus, we explored the transitions in the genetic structure of Bronze Age populations from the South-Caucasus, including Kura-Araxes individuals from Kalavan-1, the Talin necropolis and the tombs of Kaps in Armenia[1,36] as well as all the individuals recently published by Lazaridis et al.[22]. Sadly, no ancient DNA could be retrieved from the Mentesh Tepe Chalcolithic levels. With almost all D-stats of the form D(Mbuti, Mentesh; Caucasus BA, Anatolia BA/Caucasus BA) being null, we do not see any preferential gene flow from Mentesh Tepe into one Bronze Age population from the South Caucasus or Anatolia (Supplementary Fig. 6).

The PCA shows that all the Bronze Age individuals from Armenia plot together and are shifted toward the Steppe cluster. In the ADMIXTURE analysis, they all exhibit a red component, absent in the Neolithic Mentesh Tepe individuals but maximised in Steppe populations and present, also, in CHG individuals. Interestingly, individuals from Chalcolithic Armenia (from Areni-1 cave, four of whom are directly dated by C14) do carry this Steppe/CHG component, whereas a Chalcolithic individual from Alkhantepe in Azerbaijan does not. D-statistics of the form D(Mbuti, Steppe Eneolithic; Mentesh Tepe, South Caucasus Bronze Age) are almost all significantly positive (Z-score: +2.1 to +5.8), highlighting a gene flow to the South Caucasus from the Steppes or from a population linked to CHG after the Neolithic period. This result can be interpreted with two different hypotheses: either a Neolithic population from North-Caucasus or an ancestral population from the South Caucasus carrying a small proportion of Steppe/CHG ancestry replaced the local Mentesh-like Neolithic population in South-Caucasus, or a population from the steppe north of the Caucasus migrated south and admixed with the local population.

To test for these hypotheses, we used qpAdm with the rotating method[37] and modelled the Chalcolithic and Bronze Age populations found in Armenia. The only fitting model for Areni-1 cave (Chalcolithic Armenia) is an admixture between 25% Steppe and 75% Mentesh ($p$-value = 0.02) (Supplementary Fig. 7). During the Bronze Age, we observe an increase in Steppe contribution from the Early Bronze Age Kura-Araxes (0–10% Steppe contribution) to the Middle and Late Bronze Age individuals (around 40% Steppe contribution). This increase could be linked to a wave of migration from the north during the Bronze Age, or to a continuous admixture between Steppe and South Caucasus populations, maybe through North-Caucasus groups, as the latter are genetically close to the South Caucasus population during the Maikop period/Bronze Age[36]. For the Early Bronze Age populations of Armenia, we also note that the best models ($p = 0.08$ for Talin and Karavan and $p = 0.51$ for Kaps) are involving CHGs instead of a Steppe population. This suggests that the admixture at the base of the Kura-Araxes ancestry occurred between an unsampled population from the Caucasus with a profile more similar to the Caucasus Hunter-Gatherers and a Mentesh-like population from the Late Chalcolithic period. Thus, it is unlikely that Kura-Araxes populations had yet received a significant gene flow from the Steppe at this period. Models involving the only Middle Chalcolithic individual from the North Caucasus (from the Unakozovskaya cave)[36] available to date did fit but not as well as the other models ($0.01 < p$-value < 0.05). Though we note that for Kura-Araxes the models involving the North Caucasus individuals were more successful than for the LBA populations, no nested model involving 100% of North Caucasus Neolithic was detected. In other words, the Bronze Age or Chalcolithic individuals found in Armenia cannot be modelled as 100% North-Caucasus Neolithic. This result suggests that the admixture scenario is more likely than the migration one, given the individuals sequenced for now.

**Kinship**. To establish the kinship structure and social organisation of the four sequenced Neolithic Mentesh Tepe individuals, we first look at uniparental markers. Three different mitochondrial haplogroups were determined: U7, K1 and U1a1. Both MT23 and MT26 shared the same mitochondrial haplogroup, as well as the same Y chromosome haplogroup (J2b).

Next, genetic relatedness analyses were performed to further investigate potential close familial ties within individuals of structure 342. To this intent and given the low coverage of our samples, kinship analyses were carried out using READ[38].

Although READ can infer relatedness from extremely low-coverage data, a limitation of this method is that it generally requires a cohort of input individuals. It is based on the assumption that most of them are genetically unrelated to compute a sound estimate of the genetic diversity found within our population of interest. To overcome this issue, we extracted the genotype of the Neolithic MTT001, along with 21 previously published Anatolian individuals from the Late Chalcolithic/Early Bronze-Age Arslantepe (ART) and Tell Kurdu from both Neolithic and Early Chalcolithic period[23] (Fig. 2a). Here, the intent is to use these individuals as a proxy population, to calculate the median of the average pairwise distances between individuals ($\overline{P}0$), while minimising the risk of introducing biases arising from a mismatch between the genetic diversity found within burial 342 and that of our surrogate population.

Using this approach, we detected elevated levels of genetic relatedness between individuals MT23–MT26, with a non-normalised average pairwise SNP mismatch ratio of P0 0.239. This value equates to $\overline{P}0$ 734 ± 0.024 when normalised over the median of ART individuals, which can be confidently translated as first-degree individuals (Fig. 3, Supplementary Fig. 8 and Supplementary Data 5), and with a high number of overlapping

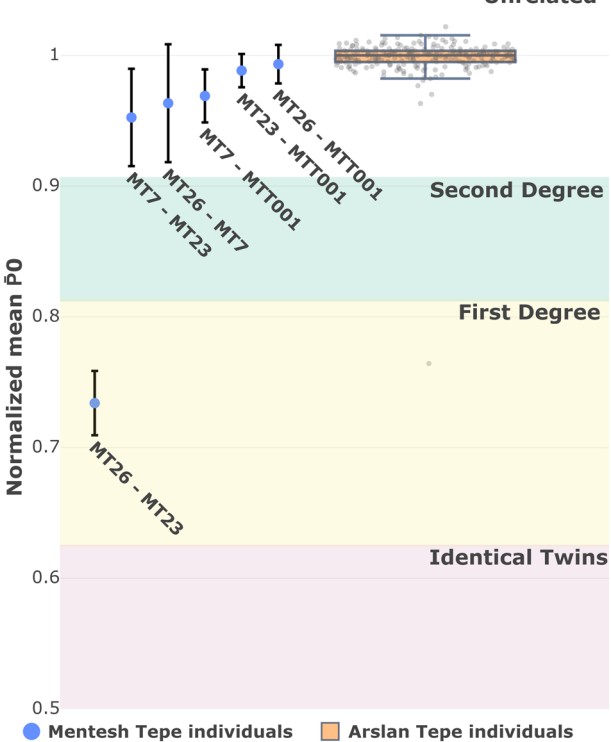

**Fig. 3 Estimation of genetic relatedness of newly and previously sequenced Mentesh Tepe individuals.** Sorted normalised average P0 values for all pairwise comparisons between Mentesh Tepe individuals (blue markers) and Arslan Tepe Late Chalcolithic/Early Bronze-Age Arslantepe individuals (orange boxplot), using the READ estimation method. Marker error bars indicate 95% confidence intervals for the observed average P0 estimation. Coloured areas represent the thresholds at which relatedness orders are considered by READ. Boxplot highlights the distribution of the observed normalised P0 of the Arslan Tepe individuals used for the normalisation step of READ, with each point corresponding to a given pair of proxy individuals (jittered along the horizontal axis). Lower and upper fences respectively represent the minimum and maximum sample point, excluding observations located below or above 1.5 the interquartile range. Boxspan: 25th–75th percentile.

SNPs ($n = 17864$) which is well over READ's satisfactory threshold of around 1500 SNPs. These results were subsequently confirmed using TKGWV2[39], an alternative kinship estimation method which provided us with the same findings (Supplementary Fig. 9).

However, close genetic ties could not be detected regarding the other pairs involving individuals MT7 and MTT001. The pairs MT23–MT7 and MT26–MT7 were both estimated with a normalised value neighbouring the observed median of ART individuals ($\overline{P0}$ 0.952 ± 0.037 and $\overline{P0}$ 0.963 ± 0.045, respectively, Supplementary Data 5). However, READ's inability to detect any order of kinship greater than the second degree and the high uncertainty in these latter results should be considered when interpreting them, as is demonstrated by the high amplitude of the associated confidence intervals and given the low number of overlapping SNPs between these individuals and MT7 ($n = 8674$ and $n = 5561$ for MT23–MT7 and MT26–MT7, respectively).

## Discussion

Our genetic kinship analysis results regarding pair MT23–MT26 unequivocally indicate that these two individuals share a first order of relatedness. This observation, combined with their close

age of death, their shared mitochondrial haplogroups, and the context in which the remains of these two individuals were discovered within structure 342, is strongly consistent with the hypothesis that they are, in fact, siblings. It should be noted that the degree of relationship estimated for this pair remained consistent throughout subsequent attempts involving different surrogate populations, including Late Neolithic individuals of Tell Kurdu (Supplementary Fig. 8).

Even after removing MT26, as he is the brother of MT23, the other samples, MT23, MTT001, and MT7, show strong genetic homogeneity despite the different sequencing approaches between MTT001 and the others. However, no nuclear or mitochondrial genetic relationships have been observed between these individuals. This suggests that Mentesh Tepe represents a highly homogeneous population or an extended family uniting people related by more than 3° or more. This observation is not surprising considering the high number of individuals in the structure and the fact that they do not represent the whole community: indeed, adult males are underrepresented, and the bodies have been buried by people aware of the family relationships between them (as exemplified by the staging of MT23 and MT26).

On a regional scale, a certain degree of homogeneity is observed. The Polutepe group, though belonging to a different Neolithic culture, is highly similar to that of Mentesh Tepe, as is the Masis Blur Neolithic group, belonging to the Aknashen culture, a variant of the Shomu-Shulaveri culture. While the Aknashen individual is more Caucasian-like than the other South Caucasus Neolithic individuals, this does not constitute a significant difference, showing that a coherent culture displays a small genetic heterogeneity. Moreover, this period is characterised by several other archaeological cultures, such as Kültepe or Kamiltepe, that have not been studied yet through paleogenetic studies. As the Shomutepe-Shulaveri is already not completely homogenous, it would not be surprising that the population linked to these cultures are genetically different. In addition, the lack of archaeological background for Polutepe prevents us from drawing any conclusions about the associated culture. However, we do show here that the Aknashen culture presents a good degree of genetic homogeneity with the Shomu-Shulaveri individuals but is not fully parallel to the overall cultural unity Aratashen-Shomu-Shulaveri culture seen by the archaeologists.

The genetic analyses performed on the Mentesh Tepe samples allow a better understanding of the diffusion of the Neolithic in this area. Based on the material culture, two models were considered: after a very short transitional phase of an aceramic culture, the Neolithic package appeared with the SSC, either from acculturation or from the migration of human groups from Western Asia[9]. The genetic data from Mentesh Tepe shows that this SSC site shared one ancestry probably inherited from the Mesolithic Caucasian Hunter-Gatherers (or the Neolithic Iranian Farmers), one from the early Anatolian farmers and one from North-eastern Mesopotamia. The genetic proximity to southern populations could be an indication of the role that Halaf and Hassuna communities played in the formation of the SSC, as already identified through the material culture. Due to the poor DNA preservation, genetic data from the core of Mesopotamia remains scarce, and the genetic affinity of Halaf groups can only be determined by their proximity to South-eastern Anatolia and North-eastern Mesopotamia. Interestingly, Tell Kurdu, which has known a strong influence on the Halafian culture, appears as the best proxy for the ancestors of Mentesh Tepe individuals in the qpAdm analyses. The other part of Mentesh Tepe ancestry is best modelled by Iranian Neolithic groups from Ganj Dareh[40]. This genetic component is also shared with Caucasian and Iranian Hunter-Gatherer, suggesting that a large population inhabited the southern fringes of the Caucasus Mountains and coasts of the

Caspian Sea at the end of the Neolithic period. The currently available data could not discriminate if this ancestry was local (for instance, from the Early Neolithic groups evidenced at Chokh, Kmlo-2 or Damjili Cave) or came from Iranian farmers, thus leading to a complete replacement of the more ancient groups.

Like Skourtanioti et al.[23]. analyses, which date the admixture event to 6500 BC, we obtain an admixture event around the middle of the 7th millennium, even if their work used two individuals from two different sites with diverse backgrounds, dates and possibly genetic origin, while we focused our analyses on a homogeneous population. However, due to the high number of missing data in the populations used to calculate the date of admixture, the fit is not perfect. Mentesh Tepe appears as a site that closely followed the process of migration and admixture of Anatolian farmers and of North-eastern Mesopotamian populations with the South Caucasus earlier Hunter-Gatherers' descendants, which was contemporaneous with the establishment of the Shomu-Shulaveri culture and is coherent with the ancient dates obtained for this site.

The genetic data from Mentesh Tepe also helps understanding better the subsequent periods, as previous studies exploring the Bronze Age in the South Caucasus only had available the Chalcolithic genome-wide data from the North Caucasus[1,36].

Indeed, we find that the North Caucasus Chalcolithic individual used previously[36] as a proxy for the Chalcolithic and Neolithic Caucasus shows a genetic profile that differs from the Mentesh Tepe one, with a slight amount of Eastern Hunter-Gatherer component likely due to a gene flow from the adjacent steppe. The comparison was not directly feasible in the former study due to the lack of Neolithic data from the southern slope of the Great Caucasus, which emphasises the need for a thorough sampling in ancient DNA studies, both in space and time.

Thanks to the new data, we show that, after the end of the Neolithic, a genetic component related to the CHG reappeared in the South Caucasus and that, later in the Bronze Age, a genetic component related to the Pontic steppe groups arrived. Chalcolithic and Late Bronze Age groups in the South Caucasus can be modelled as a mixture of Mentesh Tepe-related populations and the Steppe population. Interestingly, Steppe ancestry is not spread homogeneously during the Chalcolithic period, whereas Areni-1 individuals display a high amount of steppe ancestry; none is found in the Alkhantepe genome. Such diversification in the population's structures could mirror the one observed in cultural features. The funerary practices were also more diversified during the Chalcolithic, and while the dead at Alkhantepe are still buried in pits according to ancient practices, the first kurgans are observed in Kavtiskhevi (Georgia) and Soyuq Bulaq (Azerbaijan)[41,42]. Part of the Areni-1 pottery as well as the funerary architecture and pottery from Soyuq Bulaq connect them with the Maikop tradition[43]. The pre-Kura-Araxes South-Caucasus thus appears as a mosaic of populations of different ancestry, with the intrusion of more northern populations. Nevertheless, the precise dating of one individual from Areni 1 could be uncertain (cf. supplementary text).

Compared to the Mentesh Tepe Neolithic individuals, an archaic ancestry associated with CHG re-emerges during the Early Bronze Age/Kura-Araxes period. We also observed some Steppe ancestry in two Kura-Araxes groups (Kaps, Shengavit), but at the lowest proportion than during the Late Bronze Age. At the end of the Kura-Araxes period, more frequent connections with the Steppes are also evidenced in the burial rituals[44]. This gene flow from the north mirrors the earlier (Maikop period) increase of the South Caucasus ancestries in the populations living on the northern slopes of the Great Caucasus after the end of the Neolithic, thus showing that the mountains act as a bridge rather than a frontier.

The analysis of ancient DNA from the Mentesh Tepe Neolithic funerary pit reveals an uneven genetic homogeneity inside this Shomu-Shulaveri culture archaeological site. After the dramatic event that caused their death, the two brothers were buried, embracing each other, by people knowing their close relationship. The Shomu-Shulaveri culture took place between two intense population events: a population expansion of Neolithic groups from South-eastern Anatolia and Northern Mesopotamia at the beginning of the Neolithic and diffusion of steppe populations starting with the Middle of the Bronze Age period and following the resurgence of the CHG ancestry during the Kura-Araxes period. Even if the genetic analyses presented here help to better understand the timing and direction of these events, a more detailed analysis of other samples from the South Caucasus is called for, as this area is characterised by a high cultural diversity during the Neolithic period.

## Methods

**Archaeological sampling.** We obtained ethical authorisation from the Institute of Archaeology and Ethnography of Azerbaijan to study samples from Mentesh Tepe. Extraction was attempted on the 30 samples found in Structure 342 of the site, except MTT001 (Individual 1), which has already been published elsewhere[23]. Petrous bone fragments from 23 among 30 individuals of the SSC structure S342 have been selected (Supplementary Data 1) for ancient DNA analysis based on overall bone preservation.

**Ancient DNA extraction.** All pre-amplification steps were carried out in the clean room dedicated to ancient DNA of the Paleogenomic and Molecular genetics platform, set in the Musée de l'Homme (Paris). As in ref. [45], ancient DNA extraction was performed using a protocol adapted from ref. [46]. Briefly, 50–200 mg of petrous bone was powdered by drilling and incubated in 1 ml of lysis buffer (0.45 M EDTA, 10 mM Tris-HCl (pH 8.0), 0.1% SDS, 65 mM DTT, and 0.5 mg/ml proteinase K), at 37 °C, for 14 h. After centrifugation, 1 ml of supernatant was recovered and purified with 13 ml of binding buffer (5 M GuHCl, 40% 2-propanol 0.05% Tween20, 90 mM sodium acetate 2 M, 1x Phenol Red). The mixture was then transferred to a High Pure Extender Assembly column (Roche High Pure Viral Nucleic Acid Large Volume Kit) and centrifuged. Then, the column was washed using manufacturer's recommendations (briefly, 500 µl of inhibitor removal buffer, centrifugation, twice 450 µl of wash buffer). DNA was eluted in 100 µl of elution buffer. First, we tested the presence of human ancient DNA through PCR by targeting a portion of the mitochondrial genome using the same protocol as in ref. [45]. Only samples with positive mitochondrial DNA amplification were used for library preparation.

**Library preparation.** The ancient DNA extract was converted to a TruSeq Nano Illumina library using the manufacturer's protocol with slight modifications that account for the ancient DNA damage. First, DNA was not fragmented for obvious reasons; only 25 µl of ancient DNA extract was used; after end-repair, the libraries were purified using a MinElute column (Qiagen ©); the adaptor mix was gently pre-heated before adding ligase enzyme; libraries were amplified using 10–12 PCR cycles and purified on a MinElute column (Qiagen ©). Analyses on a LabChip ® GX provided an estimated size distribution of fragments with a peak length of 150–250 bp. Due to the limited preservation of genetic material in the site, reliable ancient DNA was found only in seven individuals, and genome-wide data were generated only for three of them.

For the three selected samples (endogenous DNA content >1%), genomic capture was performed using the myBaits Expert Whole Genome Enrichment (WGE) kit (Arbor Biosciences), and the manufacturer's instructions were followed. Baits were formed from the genomic DNA of three individuals of different (African, European and Asian) ancestries.

**Sequencing.** Aliquots of the amplified libraries were first pooled and sequenced on MiSeq instrument (2 × 75 bp) on the IGenSeq platform. Captured libraries were sequenced on a NextSeq 500 (2 × 75 bp) instrument.

**Data processing.** Considering that our kinship analysis protocol makes use of previously published ancient individuals as a surrogate population, a separate data-preprocessing workflow mirroring that of ref. [23] was devised to make our results as comparable as possible with these previous analyses. All of our newly sequenced samples were thus specifically preprocessed through EAGER-v1.92.37[47]. From the raw shotgun sequencing data, sequencing adaptors were thus clipped, and paired-end reads were merged using AdapterRemoval-v2.2.1a[48], all the while discarding any read shorter than 30 bp. Reads were then aligned against the 1000Genomes-phase2 reference genome (hs37d5) using BWA-v0.7.17-r1188[49], with a quality threshold for read trimming of q30. Samples were then merged across sequencing

runs for each individual, using samtools merge v1.6.0[50]. PCR duplicates were marked and removed using DeDup v0.12.1[47] prior to and following the previously described merging. PMD-damage statistics and frequency of nucleotide mis-incorporation were then estimated until the 50th nucleotide (starting from the 3′ and 5′ ends) using MapDamage v2.0.6[51] (Supplementary Fig. S1). Quality scores within the bam files were then rescaled accordingly with the latter software, using default parameters. We extracted reads overlapping known variants from the 1240K dataset from the rescaled bam file using samtools mpileup, while disabling per-Base Alignment Quality (-B flag) and skipping any alignment carrying a base and/or mapping Phred-quality score which was lower than 30 (−q and −Q flags). Random pseudo-haploid variant calling was then performed using SequenceTools PileupCaller-v1.5.0 (Download: https://github.com/stschiff/sequenceTools). To evaluate contamination, we used AuthentiCT[52], and for the male samples, we calculated contamination using the X chromosome[53] with ANGSD[54]. (Supplementary Data 1).

**Merging genomic data.** We selected 3529 published ancient human genomes from Eurasia (Supplementary Data 2) from the Palaeolithic to Middle Age, whom DNA sequencing data were generated with whole genome shotgun or hybridisation capture technics from the merge dataset v42.4 available at https://reich.hms.harvard.edu/allen-ancient-dna-resource-aadr-downloadable-genotypes-present-day-and-ancient-dna-data and in two recent paper about the South Caucasus and Anatolia[22,23]. We retained the nonrelated individuals with more than 10,000 SNPs hit on the 1240k panel. We analysed ancient data with 1587 Eurasian individuals from the Human Origin dataset. We merged our genome-wide data to this dataset using *mergeit* from the EIGENSOFT package suite[55], and we haploidised them by randomly selecting one allele per position. The final merge includes 597,573 SNPs for 5116 individuals, and we call it the HO-dataset. For the analysis requiring more SNPs, we use a merge without the modern individuals from the HO dataset, covering 1,233,013 SNPs, which we call the 1240k dataset.

**PCA.** We ran PCA with smartpca[55] using the HO-dataset, on 1390 European and Middle Eastern individuals, and we projected all the ancient samples. We used the default parameters with lsqproject: YES, and numoutlieriter: 0 settings.

**ADMIXTURE.** We computed ADMIXTURE[33] analysis using the HO dataset, where all populations were downsampled to a maximum of 20 individuals. We performed the analysis on 1266 Eurasian modern individuals, including East Asian, and 2526 ancient samples from the first dataset on a subset of 365,075 SNPs pruned for linkage disequilibrium (by using PLINK --indep-pairwise 200 25 0.4 function). We ran 20 replicate ADMIXTURE analyses for K between 2 and 14, from which we kept the most likely.

**D-statistics.** We performed D-statistics on the 1240k dataset using the qpDstat program of the ADMIXTOOLS package[56] with Mbuti as outgroup for both statistics. We computed all the D-statistics for each Mentesh Tepe individual and for the group Mentesh Tepe formed by MT23, MT7 and MTT001. We computed D-statistics of the form D(Mbuti, Y; Z, MT), with Z being Palaeolithic or Neolithic groups from Anatolia (Barcın site in North Anatolia, TellKurdu site from Southeastern Anatolia and Tepecik site from Central Anatolia), Northern Mesopotamia (Nemrik 9 and Shanidar sites from Irak, Mardin site from Turkey) or Caucasus (CHG, Polutepe, Armenia Neolithic, Armenia Chalcolithic, North Caucasus Chalcolithic) and Y ancient Palaeolithic, Mesolithic or Neolithic populations from Western Eurasia to test the affinity of Mentesh Tepe to them when compared to an older contemporaneous group from the same region. We also performed D-stat of the form D(Mbuti, MT; Anatolia or Caucasus BA, Caucasus BA) to test the affinity of Mentesh Tepe with the populations that succeeded it in the region. These last results are shown in the supplementary information (Supplementary Fig. 3). All the results of the D-statistics analyses, including the data used to generate the figures, are in Supplementary Data 3.

**qpAdm analysis.** We performed rotating qpAdm analysis from ADMIXTOOLS package[56], using the 1240k dataset, to model the ancestry of Mentesh Tepe and Polutepe populations. We used Mbuti.DG, Ami.DG, Mixe.DG, Russia_Kostenki14.SG, Russia_MA1_HG.SG, EEHG, Italy_North_Villabruna_HG, Natufian as reference populations. Prior to the analysis, we checked if the reference populations could discriminate well between the source populations by computing qpWave.

We tested a rotating group with Iran_GanjDareh_N, CHG, Russia_Caucasus_Eneolithic, PPN, Barcın_N, Anatolia_TellKurdu_LN IRQ_Nemrik9_PPN and TUR_SE_Mardin_PPN to assert the best model of a two-way admixture for each of our individuals. We then run the same qpAdm analysis for the Mentesh group form by MT23, MT7 and MTT001 and all the South-Caucasus Neolithic (SCN) individuals in one group. We also tested for 3-way admixture models; we could obtain model without nested 2-way models for Mentesh. All the results of the qpAdm modelling, including the data used to generate the figures, are in Supplementary Data 4.

**DATES.** We used DATES v75318[35] to estimate the time of admixture events between Iran_N and Anatolia_N in the Mentesh group. To convert the estimated admixture date in a generation into years, we assumed 29 years per generation[13]. The standard errors of DATES estimates come from the weighted block jackknife with "binsize: 0.001", "maxdis: 1", "runmode: 1", "mincount: 1", "lovalfit: 0.45" as parameters as in the example file at https://github.com/priyamoorjani/DATES/blob/master/example/par.dates. We tested three different configurations to account for the poor coverage of the ancient individuals: the first test was conducted with only ancient Iranian Neolithic individuals as a source and Barcın_N or Tell Kur-du_LN as the second source. The second, with modern populations from Armenia, Georgia and the Russian Caucasus added in the Iranian source. A third and final test was made with Barcın_N and Tell Kurdu_LN individuals put together as a source. We performed these three configurations for the Mentesh individuals grouped together and for all the Neolithic South-Caucasus individuals as a group.

**Uniparental markers.** Y chromosome and mitochondrial haplogroups were determined by comparison of vcf files to databases. For mitochondrial DNA, a comparison was performed on Phylotree 17, using Haplogrep v2.2;[57] for Y chromosome, the comparison on ISOGG version July 11, 2020, was done manually.

**Kinship.** Using PLINK-v1.90b6.16[58,59], we first filtered out from our 1240K dataset anything but biallelic autosomal SNP positions carrying a minor allele frequency greater or equal to 5% and subsequently extracted the genotypes of individuals MT23, MT26, MT7 and MTT001, along with 21 *unrelated* Late Chalcolithic and Early Bronze-Age individuals from Arslantepe (ART). The resulting dataset was then transposed to the tped/tfam format and used as input for READ. Next, we devised a two-pass estimation approach by running the READ method twice: A first run was therefore performed using only the Arslantepe individuals and with default parameters to compute a normalisation value (*normalisation method:* median; nonoverlapping window-size: 1000000). A second run was then carried out against the Mentesh Tepe individuals alone, using the previously obtained median P̄0 as a predefined normalisation value. We re-applied this protocol a second time using six individuals from the Late Neolithic and Early Chalcolithic site of Tell Kurdu to substantiate our initial results (Supplementary Fig. 8).

We also used TKGWV2 to substantiate our initial results (Supplementary Fig. 9). As per recommendations from the authors, the final BAM files from all Mentesh individuals were downsampled to around 1.8M reads beforehand[39]. We then performed the analysis using the 22M nonfixed biallelic variant set provided by Fernandes et al.[39]. To assess confidence in the raw relatedness coefficients given by the method, we obtained posterior probabilities by simulating distribution curves for each tested pair of individuals (n = 6000 simulation replicates).

**Statistics and reproducibility.** The dataset used for PCA and ADMIXURE includes 597,573 SNPs for 5116 individuals, and the dataset used for Dstats, qpAdm and DATES is covering 1,233,013 SNPs for 5,116 individuals. When an analysis targets Mentesh as a group, it means three individuals grouped together (MT7, MT23 and MT001). ADMIXTURE analysis has been replicated 20 times. For Kinship analysis, READ analysis has been replicated twice: one with 21 individuals from Arslantepe and one with six individuals from Tell Kurdu.

**Reporting summary.** Further information on research design is available in the Nature Portfolio Reporting Summary linked to this article.

## Data availability

The dataset generated for this study (bam file) can be found in the European Nucleotide Archive under accession number PRJEB54894. The data used to generate Fig. 2c are in Supplementary Data 3, for Fig. 2d in Supplementary Data 4 and for Fig. 3 in Supplementary Data 5.

## Code availability

Code to reproduce the analysis and the figure of this paper can be obtained here: https://doi.org/10.5281/zenodo.7707671, or here: https://github.com/pguarinovignon/Mentesh.git.

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

## Acknowledgements

The authors express their gratitude to the French-Azerbaijani excavation team, who contributed to the sampling. The excavation in Mentesh Tepe is partly supported by the

Ministry of Foreign and European Affairs and the ANR projects "Ancient Kura" and "Kura in Motion", directed by B. Lyonnet and B. Helwing. This project has been funded by the ANR project "Kura in Motion" and the Sorbonne University Emergence grant (PaleOxus). P.G.V. was funded by a PhD grant given by ENS de Lyon. We thank the Paleogenomic and molecular genetics platform (P2GM) of the MNHN (Paris) as well as the IGenSeq platform of the Institut du Cerveau et de la Moelle (Paris). We thank Nina Marchi for the comments and critics and Lasse Vinner for help in setting up a first version of the DNA extraction protocole.

## Author contributions

C.B. and B.L. designed the study. B.L., F.G., E.J. and L.P. provided archaeological material and advised on the archaeological background. A.C. and A.M. performed the laboratory work. P.G.V. and M.L. performed the data analyses, with C.B. providing guidance. P.G.V., M.L., B.L. and C.B. wrote the manuscript with input from all co-authors.

## Competing interests

The authors declare no competing interests.
