## [Peer Review File · Communications Biology]

Reviewers' comments:

Reviewer #1 (Remarks to the Author):

The paper " Genome-wide analysis of a collective grave from Mentesh Tepe, one of the early Shomu-Shulaveri sites in South Caucasus" presents here genomic data for three new individuals from Mentesh Tepe in Azerbaijan with the aim to shed light on the origin and population structure of the Shomu-Shulaveri community. The topic is very interesting, the analyses are very well conducted and the paper is well written.

The paper contributes to adding knowledge to Shomu-Shulaveri culture and about migration events that took place in the Neolithic period in the southern Caucasus. It is very interesting also the discovering of two brothers buried embracing each other.

I have a few things to highlight:

1. l. 118 and discussion from line l.275, please see Lugli et al., 2019 Enamel peptides reveal the sex of the Late Antique 'Lovers of Modena' and the references therein, about similar double burials
2. l.121 from this sentence it is not clear if the authors analyzed a subset of samples available, or instead if they screened all the samples, but only a few showed sufficient endogenous / coverage
3. l. 130 in Table S1 check the correspondence of coverage values used in the main text. Moreover, in this table add the values of endogenous instead of "low".
4. l.139 is probably Table S2?
5. l. 240 Kinship analysis: have you tried to use the method implemented by Fernandes et al., 2021 in the paper: "TKGWV2: an ancient DNA relatedness pipeline for ultra-low coverage whole genome shotgun data"?
6. l. 377 is not clear if you extracted DNA from all samples (30) or from the ones from which it was possible to obtain petrous bone (25). In table S1 only 24 samples are listed. Are MT16 and M16 referring to the same individual? In this way, the results of only 23 samples are here reported. Please clarify this in the main text and check the table. Moreover, if you collected petrous bones from 25 individuals, what have you sampled for others? it would be useful to specify this also in Table S1 for each individual
7. l.394 How did you test the presence of ancient DNA in the samples? Because this sentence is placed before the "Library preparation" paragraph. It seems that you used a DNA quantification like Qubit fluorometer or something similar.

Reviewer #2 (Remarks to the Author):

The authors present genetic data and analysis on three Neolithic individuals from Mentesh Tepe in Azerbaijan belonging to the beginnings of the Shomu-Shulaveri culture (SSC). This is a useful addition to the genetic data from the South Caucasus region which includes two Neolithic individuals from Azerbaijan (both from Mentesh Tepe and from Polutepe by Skourtanioti et al. 2020) and from Armenia (recently published data from Aknashen and Masis Blur by Lazaridis et al. 2022 that probably appeared before the analysis presented here but would be useful to re-analyze in a re-submission).

I would like to see in a resubmission a co-analysis of all these individuals together: are there differences between the newly reported data and those of the earlier studies of the South Caucasus Neolithic? How do the new data from Azerbaijan differ from those of Armenia to the west and from those published by Skourtanioti et al.?

The finding of two juvenile brothers at the site would be of interest to archaeologists.

The study claims that there is steppe ancestry in the Early Bronze Age Kura-Araxes populations. However, Armenia_EBA is modeled as a mixture of Anatolian Neolithic and CHG in Lazaridis et al. (2016) Fig. 4a ("Genomic insights into the origin of farming in the ancient Near East") and Fig. 5B (Lazaridis et al. 2022, "The genetic history of the Southern Arc: A bridge between West Asia and Europe")

It seems that in the paper models fit the Armenia_EBA with Late Chalcolithic Armenia ancestry plus either CHG or Steppe_Eneolithic, so it is not clear if the authors think that there is indeed steppe ancestry in EBA Kura-Araxes Armenia as opposed to increased CHG ancestry only. It would be interesting if the authors could fit a 3-way model and thus estimate separately contributions of ancestry from the pre-Kura-Araxes Chalcolithic, the Caucasus, and the Steppe.

Another area of potential interest would be Mesopotamia from which the recently published individuals (Lazaridis et al. 2022) could be useful in distinguishing the origins of the Neolithic population at Mentesh Tepe, together with data from Anatolia (Catal Hoyuk and other ancient Anatolian Neolithic farmers). It would be great if the authors can study the origins of the Neolithic in the South Caucasus in terms of the possible (Anatolian, Mesopotamian, Levantine, Zagros) origins.

Overall, I think this is very interesting data that definitely adds 3 new individuals to the 4 published from the South Caucasus Neolithic, but I would like to see some more analysis that emphasizes what - if anything - is new here and clarifies the questions I outlined above.

Fig. 2: Caucase should be Caucasus; also in a couple places in the text

Response to reviewers

Reviewer #1:

The paper “Genome-wide analysis of a collective grave from Mentesh Tepe, one of the early Shomu-Shulaveri sites in South Caucasus” presents here genomic data for three new individuals from Mentesh Tepe in Azerbaijan with the aim to shed light on the origin and population structure of the Shomu-Shulaveri community. The topic is very interesting, the analyses are very well conducted and the paper is well written.

We thank the Reviewer #1 for their kind comments on our work.

The paper contributes to adding knowledge to Shomu-Shulaveri culture and about migration events that took place in the Neolithic period in the southern Caucasus. It is very interesting also the discovering of two brothers buried embracing each other.

I have a few things to highlight:

1. l. 118 and discussion from line l.275, please see Lugli et al., 2019 Enamel peptides reveal the sex of the Late Antique ‘Lovers of Modena’ and the references therein, about similar double burials
2. l.121 from this sentence it is not clear if the authors analyzed a subset of samples available, or instead if they screened all the samples, but only a few showed sufficient endogenous / coverage

First, we tested the samples by PCR and eliminated those with Ct > 35 cycles ; then, we prepared libraries for a genomic screening, that allow determining genetic sex for those with more than 20 000 reads aligned on the human genome. And we pursue the analysis on samples with endogenous DNA content > 1%.

3. l. 130 in Table S1 check the correspondence of coverage values used in the main text. Moreover, in this table add the values of endogenous instead of “low”.

We clarify by replacing « low » by « PCR negative » or the endogenous DNA content observed after screening if the preliminary results passed the threshold.

4. l.139 is probably Table S2?

Corrected l.150

5. l. 240 Kinship analysis: have you tried to use the method implemented by Fernandes et al., 2021 in the paper: “TKGWV2: an ancient DNA relatedness pipeline for ultra-low coverage whole genome shotgun data”?

We performed this analysis and found similar results as the one presented in the manuscript. We had the results l. 325-327 and l. 611-617 and in supplementary figure 9.

6. l. 377 is not clear if you extracted DNA from all samples (30) or from the ones from which it was possible to obtain petrous bone (25). In table S1 only 24 samples are listed. Are MT16 and M16 referring to the same individual? In this way, the results of only 23 samples are here reported. Please clarify this in the main text and check the table. Moreover, if you collected petrous bones from 25 individuals, what have you sampled for others? it would be useful to specify this also in Table S1 for each individual

Thank you for identifying the line MT16 as duplicated. We analysed only the samples for which a petrous bone was preserved, with a sufficient overall preservation (23 samples). We updated the Table S1 accordingly.

7. 1.394 How did you test the presence of ancient DNA in the samples? Because this sentence is placed before the “Library preparation” paragraph. It seems that you used a DNA quantification like Qubit fluorometer or something similar.

As in Guarino-Vignon et al. 2022, we amplified by PCR a short, conserved region of the mt DNA, and only analyse samples that allowed a PCR amplification. Several tests in the lab have shown that samples that are PCR negative systematically have less than 0,01% of endogenous DNA.

Reviewer #2 :

The authors present genetic data and analysis on three Neolithic individuals from Mentesh Tepe in Azerbaijan belonging to the beginnings of the Shomu-Shulaveri culture (SSC). This is a useful addition to the genetic data from the South Caucasus region which includes two Neolithic individuals from Azerbaijan (both from Mentesh Tepe and from Polutepe by Skourtanioti et al. 2020) and from Armenia (recently published data from Aknashen and Masis Blur by Lazaridis et al. 2022 that probably appeared before the analysis presented here but would be useful to re-analyze in a re-submission).

I would like to see in a resubmission a co-analysis of all these individuals together: are there differences between the newly reported data and those of the earlier studies of the South Caucasus Neolithic? How do the new data from Azerbaijan differ from those of Armenia to the west and from those published by Skourtanioti et al.?

We already did compare the individuals by Skourtanioti et al. 2020 in the original manuscript. We added all the South Caucasus individuals from Lazaridis et al. 2022 in our dataset and performed all the population genetic analysis with the new dataset. All figures have been updated to account for the new individuals as well as the text (mainly lines 197-198, 358-361). We show that Neolithic individuals from Armenia are very similar to Azerbaijan individuals from the same period, without noticeable difference in the gene flow from Anatolia.

The finding of two juvenile brothers at the site would be of interest to archaeologists.

The study claims that there is steppe ancestry in the Early Bronze Age Kura-Araxes populations. However, Armenia_EBA is modeled as a mixture of Anatolian Neolithic and CHG in Lazaridis et al. (2016) Fig. 4a ("Genomic insights into the origin of farming in the ancient Near East") and Fig. 5B (Lazaridis et al. 2022, "The genetic history of the Southern Arc: A bridge between West Asia and Europe")

We have re-modelled all Bronze Age populations available, including the one in Lazaridis 2022. Indeed we obtain models with only CHG and South Caucasus Neolithic for Kura-Araxes and Early Bronze Age individuals. We note that we still have model with Steppe group working, even though they are rare. We still obtain very clear models with Steppe group for Late Bronze Age groups from Armenia. We updated the text (result and discussion section) in this sense (mostly lines 269-289, 416-420, 432-435).

It seems that in the paper models fit the Armenia_EBA with Late Chalcolithic Armenia ancestry plus either CHG or Steppe_Eneolithic, so it is not clear if the authors think that there is indeed steppe ancestry in EBA Kura-Araxes Armenia as opposed to increased CHG ancestry only. It would be interesting if the authors could fit a 3-way model and thus estimate separately contributions of ancestry from the pre-Kura-Araxes Chalcolithic, the Caucasus, and the Steppe.

We performed the 3-way models as requested. Very few BA populations gave us significant models ($p > 0.05$) so we did not discuss it in the main text. We added the models in Supplementary Table S4.D.

Another area of potential interest would be Mesopotamia from which the recently published individuals (Lazaridis et al. 2022) could be useful in distinguishing the origins of the Neolithic population at Mentesh Tepe, together with data from Anatolia (Catal Hoyuk and other ancient Anatolian Neolithic farmers). It would be great if the authors can study the origins of the Neolithic in the South Caucasus in terms of the possible (Anatolian, Mesopotamian, Levantine, Zagros) origins.

We included the new Near East individuals from Lazaridis et al. 2022 and performed all our population genetic analysis with them (lines 170-175, figure 2). We notably obtained a three populations models for the Mentesh group involving the Mesopotamian individuals sequenced in Lazaridis et al. 2022, highlighting the complexity of population movements at this period. Figures, text, and supplementary tables have been updated (see lines 222-229 for the models with Mesopotamian population).

Overall, I think this is very interesting data that definitely adds 3 new individuals to the 4 published from the South Caucasus Neolithic, but I would like to see some more analysis that emphasizes what -if anything- is new here and clarifies the questions I outlined above.

All new analysis proposed here have been done and the text has been generally clarified. We find a clear homogeneity in the Neolithic South Caucasus, and a new model involving CHG, Anatolian farmers and Mesopotamian farmers for the Mentesh group.

Here the figure 2 (next page) and supp fig 4 and 6 that show the re-analyses done with the new dataset

Fig. 2: Caucase should be Caucasus; also in a couple places in the text

All “Caucase” have been replaced by “Caucasus”

Figure S4

Figure S6

REVIEWERS' COMMENTS:

Reviewer #1 (Remarks to the Author):

The authors have complied with all the comments raised by the reviewers and I have no other suggestions.

Reviewer #2 (Remarks to the Author):

The authors have addressed most of my concerns and I recommend publication. I hope that a few remaining ones below can be addressed in the final form of the paper:

- Aknashen and Masis Blur should be marked in Fig. 1A so that the reader will have a complete picture of the geography of South Caucasus Neolithic samples.

- In the PCA (panel A) the Mentesh Type individuals are marked but the other samples from the South Caucasus Neolithic are not. All South Caucasus Neolithic samples -most relevant to this study- should be clearly marked. Panel D should also show models for them (not only for Mentesh). I believe there is enough space for them; if not, the D-statistics could be relegated to supplementary figures. I think the models of Mentesh and related populations are more important than the statistics of panel C.

- "as well as all the individuals recently published by Lazaridis et al. 2022" Should probably be a numbered reference to be in accordance with rest of manuscript style.

Lines 358-361: "On a regional scale, the same degree of homogeneity is observed. The Polutepe group, 359 though belonging to a different Neolithic culture, is highly similar to that of Mentesh Tepe, 360 as is the Armenian Neolithic group (Aknashen and Masis Blur), belonging to the Aknashen 361 variant of the Shomu-Shulaveri culture."

I think this sentence needs to be clarified to show whether you mean (a) that the mentioned populations are indistinguishable/cladal, that is, you can't find meaningful differences between them, or (b) they are similar to each other in the greater West Eurasian context while also being genetically differentiated. For the Aknashen and Masis Blur individuals, their publication (<https://www.science.org/doi/10.1126/science.abq0762>) claims that:

"The two individuals from Armenia, from the sites of Aknashen (~5900 BCE) and Masis Blur (~5600 BCE) differ in being more Caucasus and Anatolia/Levant-like, respectively, despite being buried just ~200 km and a few centuries apart; thus, Neolithic people of Armenia were not homogeneous but instead exhibited variation that also encompassed two ~5700 to 5400 BCE individuals buried in neighboring Azerbaijan (7), who are intermediate between the two from Armenia in both PCA and the five-way model."

There is a seeming contrast between the finding there of non-homogeneity with your finding of homogeneity, so you should clarify whether you mean (a) or (b) and if your finding is substantial or a matter of different emphasis.

Response to reviewers

Reviewer #2 :

The authors have addressed most of my concerns and I recommend publication. I hope that a few remaining ones below can be addressed in the final form of the paper:

- Aknashen and Masis Blur should be marked in Fig. 1A so that the reader will have a complete picture of the geography of South Caucasus Neolithic samples.

The map has been updated.

- In the PCA (panel A) the Mentesh Type individuals are marked but the other samples from the South Caucasus Neolithic are not. All South Caucasus Neolithic samples -most relevant to this study- should be clearly marked. Panel D should also show models for them (not only for Mentesh). I believe there is enough space for them; if not, the D-statistics could be relegated to supplementary figures. I think the models of Mentesh and related populations are more important than the statistics of panel C.

We added a zoomed panel in the PCA to marked all the neolithic sample from the Caucasus. We added the qpAdm models in the Figure 2.D for Polutepe, Masis Blur and Aknashen individuals.

- “as well as all the individuals recently published by Lazaridis et al. 2022” Should probably be a numbered reference to be in accordance with rest of manuscript style.

Updated

Lines 358-361: “*On a regional scale, the same degree of homogeneity is observed. The Polutepe group, though belonging to a different Neolithic culture, is highly similar to that of Mentesh Tepe, as is the Armenian Neolithic group (Aknashen and Masis Blur), belonging to the Aknashen variant of the Shomu-Shulaveri culture.*” I think this sentence needs to be clarified to show whether you mean (a) that the mentioned populations are indistinguishable/cladal, that is, you can't find meaningful differences between them, or (b) they are similar to each other in the greater West Eurasian context while also being genetically differentiated. For the Aknashen and Masis Blur individuals, their publication (<https://www.science.org/doi/10.1126/science.abq0762>) claims that: “*The two individuals from Armenia, from the sites of Aknashen (~5900 BCE) and Masis Blur (~5600 BCE) differ in being more Caucasus and Anatolia/Levant-like, respectively, despite being buried just ~200 km and a few centuries apart; thus, Neolithic people of Armenia were not homogeneous but instead exhibited variation that also encompassed two ~5700 to 5400 BCE individuals buried in neighboring Azerbaijan (7), who are intermediate between the two from Armenia in both PCA and the five-way model.*” There is a seeming contrast between the finding there of non-homogeneity with your finding of homogeneity, so you should clarify whether you mean (a) or (b) and if your finding is substantial or a matter of different emphasis.

The text indeed lack precision and we would like to thanks the reviewer for the methodic remarks. The true statement is the mixture of (a) and (b). Indeed we find different percentage of CHG in Aknashen individual than in the other groups, but when tested it's not significant. The Aknashen individual does not plot with the main cluster and is drawn to the CHG cluster. So we indeed find that Aknashen is more Caucasian-like than the other South Caucasus Neolithic individuals, but it's a very slight signal and we did not want to take it to far. We updated the text to precise our findings 1.151, 1.161, 1.194, 1.213, 1.333-346.